# Citywide Cellular Traffic Prediction Based on a Hybrid Spatiotemporal Network

**Dehai Zhang \*, Linan Liu, Cheng Xie, Bing Yang and Qing Liu**

School of Software, Yunnan University, Kunming 650504, China; liulinan@mail.ynu.edu.cn (L.L.);
xiecheng@ynu.edu.cn (C.X.); yang.bing@mail.ynu.edu.cn (B.Y.); liuqing@ynu.edu.cn (Q.L.)

\* Correspondence: dhzhang@ynu.edu.cn

**Abstract:** With the arrival of 5G networks, cellular networks are moving in the direction of diversified, broadband, integrated, and intelligent networks. At the same time, the popularity of various smart terminals has led to an explosive growth in cellular traffic. Accurate network traffic prediction has become an important part of cellular network intelligence. In this context, this paper proposes a deep learning method for space-time modeling and prediction of cellular network communication traffic. First, we analyze the temporal and spatial characteristics of cellular network traffic from Telecom Italia. On this basis, we propose a hybrid spatiotemporal network (HSTNet), which is a deep learning method that uses convolutional neural networks to capture the spatiotemporal characteristics of communication traffic. This work adds deformable convolution to the convolution model to improve predictive performance. The time attribute is introduced as auxiliary information. An attention mechanism based on historical data for weight adjustment is proposed to improve the robustness of the module. We use the dataset of Telecom Italia to evaluate the performance of the proposed model. Experimental results show that compared with the existing statistics methods and machine learning algorithms, HSTNet significantly improved the prediction accuracy based on MAE and RMSE.

**Keywords:** communication traffic prediction; intelligent traffic management; deformable convolution; attention mechanism

## 1. Introduction

With the advent of fifth-generation mobile networks (5G), the cellular Internet of Things (IoT) has become a popular topic in industry [1,2]. The Groupe Speciale Mobile Association (GSMA) predicted that by 2020, the number of IoT connections will exceed 30 billion, and the number of connections based on cellular technology will reach one to two billion. The current 4G wireless network has greatly affected our lives, and the stable communication system brought by the future 5G will become a powerful promoter of Industry 4.0 [3–5]. Real-time and secure data transmission is an important guarantee for Industry 4.0, and 5G has the characteristics of large transmission data, high security, and short delay time.

At the same time, the explosive growth of global mobile devices and the IoT has also accelerated the era of big data [6,7]. Communication equipment plays an increasingly important role in people's daily lives, such as sensing, communication, entertainment, and work. A large number of communication services have generated countless mobile data; the wireless cellular networks carrying the data have become increasingly advanced and complex; and a large quantity of real-time system operation data is generated every moment. To realize intelligent management of cellular networks, it is very important to perform real-time or non-real-time regular analysis and accurate prediction of cellular traffic. For example, accurate prediction of future traffic can greatly increase the efficiency of demand aware resource allocation [8].

However, cellular network traffic changes have elusive rules, and the variation in traffic in a particular region is strongly correlated with many external factors, such as business, location, time, and user lifestyle. To extract more effectively the changing characteristics of cellular network traffic, many related studies have been carried out. The existing methods can be divided into two types: statistical or probabilistic methods and machine learning methods.

For the first kind of methods, this includes the autoregressive integrated moving average (ARIMA) [9,10], $\alpha$-stable distribution [11], and covariance function [12]. In the traffic prediction problem, these methods have comprehensively studied the characteristics of cellular networks and have shown that changes in communication traffic have both temporal autocorrelation and spatial autocorrelation. However, as the communication modes of cellular networks become more complex and subject to many external factors, these traditional linear statistical methods are not suitable for current communication traffic prediction problems.

With the development of artificial intelligence technology, machine learning methods have been widely used in industry and have also been used for cellular network traffic prediction in recent years [13]. Early researchers proposed using linear regression [14] and SVM regression [15] to predict cellular traffic. Many studies have also proposed methods for traffic prediction based on deep learning. In [16], the authors proposed a deep learning based prediction method to simulate the long term dependence of cellular network traffic in 2017. The method mainly uses self-encoded depth models and long short term memory cells (LSTM) for space-time modeling. However, the processing of self-encoding will lose some of the original information, and the ability to extract spatial features needs to be improved. In 2018, Zhang et al. [17] proposed a cellular traffic prediction method based on a convolutional neural network (STDenseNet); however, this method did not consider the impact of external conditions on traffic prediction, and the traditional convolution method had a limited effect on the complex spatial characteristics of cellular traffic.

Motivated by the aforementioned problems, based on STDenseNet, this work proposes a new hybrid spatiotemporal network (HSTNet). First, the deformable convolution unit is used in the model to improve the ability to extract complex spatial features. Then, time characteristics are introduced to enhance the accuracy of traffic prediction. Finally, an attention mechanism based on traffic history data is proposed to further enhance the robustness of the model.

## 2. Data Observation and Analysis

The wireless communication data analyzed in this paper were from Telecom Italia, which is the traffic statistics sent or received by users in specific areas of Milan [18]. The dataset consisted of a time series of traffic from 1 November 2013 to 1st January 2014, with an interval of 10 min, and included three parts: short message service (SMS), call service (Call), and Internet access (Internet). The entire urban area was divided into cells of size $H \times W$. $H$ and $W$ represent the number of rows and columns of cells. In this dataset, $H = W = 100$, indicating that the area of Milan is composed of a grid overlay of 10,000 cells with a size of about $235 \times 235$ square meters, and the value of the cell represents the statistical value of the traffic in and out of this area. Traffic data were recorded from 00:00 11/01/2013 to 23:59 01/01/2014. We merged data at ten minute intervals into hour intervals and divided each dataset into 1488 (62 days $\times$ 24 h) fragments. In the datasets, SMS and Call contained two dimensions of traffic, namely receiving and sending. The Internet only recorded the traffic that was accessed. In order to compare the spatiotemporal characteristics of the traffic of the three services clearly, we combined the traffic of the receiving and sending dimensions into one.

Thus, the entire dataset could be represented as data of $[c, t, H, W]$ dimensions $F_{c,t}$ where $c$ represents the type of cellular traffic in the dataset, $c \in \{SMS, Call, Internet\}$. $t$ represents the time interval of the flow, and $t = 1$ h in this work. $H$ and $W$ are as described above.

$$
\mathbf{F}_{c,t} = \begin{bmatrix} f_{c,t}^{(1,1)} & f_{c,t}^{(1,2)} & \cdots & f_{c,t}^{(1,W)} \\ f_{c,t}^{(2,1)} & f_{c,t}^{(2,2)} & \cdots & f_{c,t}^{(2,H)} \\ \vdots & \vdots & \ddots & \vdots \\ f_{c,t}^{(H,1)} & f_{c,t}^{(H,2)} & \cdots & f_{c,t}^{(H,W)} \end{bmatrix} \tag{1}
$$

where $f_{c,t}^{(H,W)}$ represents the traffic statistics of the cell (*H*, *W*) of the traffic data of type *c* at time *t*.

### 2.1. Temporal Domain

Figure 1 shows the trends in three different traffic flows over a 48 h period; from top to bottom, SMS, Call, and Internet. From the picture, we can clearly find that the three different traffics were subjected to a strong daily time change pattern. Basically, the traffic started to increase at approximately eight o'clock in the morning and then stayed above a very high level and started to fall at approximately eight o'clock in the evening. There were also significant differences between different traffics. The Internet traffic remained constant even at night, and Call and SMS were mostly concentrated during the day. These rules clearly correspond with the daily lives of people. Compared to daytime work and life, people make very few SMS messages and calls at night. However, the Internet not only provides contact needs, but automatic access equipment, entertainment, and other factors will lead to a large number of Internet accesses at night. Therefore, the day and night trend is relatively small compared to SMS and Call.

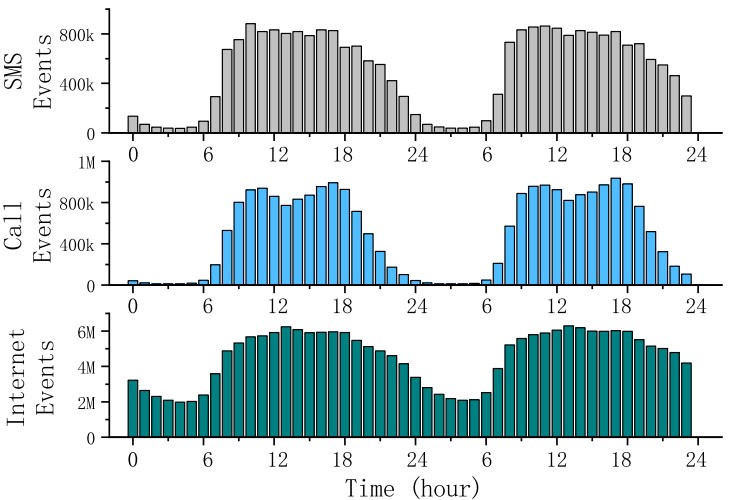

**Figure 1.** Hourly traffic change statistics.

Figure 2 is the dynamic graph of the daily total traffic in November 2013. The three types of traffic had obvious differences on working days and holidays, and the traffic on holidays was much lower than that on working days, which showed whether it was a working day that had an important impact on daily traffic. For example, the first day in the picture is an Italian legal holiday, and the second and third days are the weekend; the total traffic on these three days was significantly less than the next five working days.

Comparing the changes in the three types of traffic on weekdays and holidays, we found that the gap of Call traffic on the two dates was considerable, and the working day traffic was generally close to twice the holiday traffic. SMS traffic trends were smaller than Call, but the gap was also large. The reason was obvious: users needed to communicate more during the working day. Similar to the statistical rule of Figure 1, the daily traffic trend of the Internet was smaller than SMS and Call.

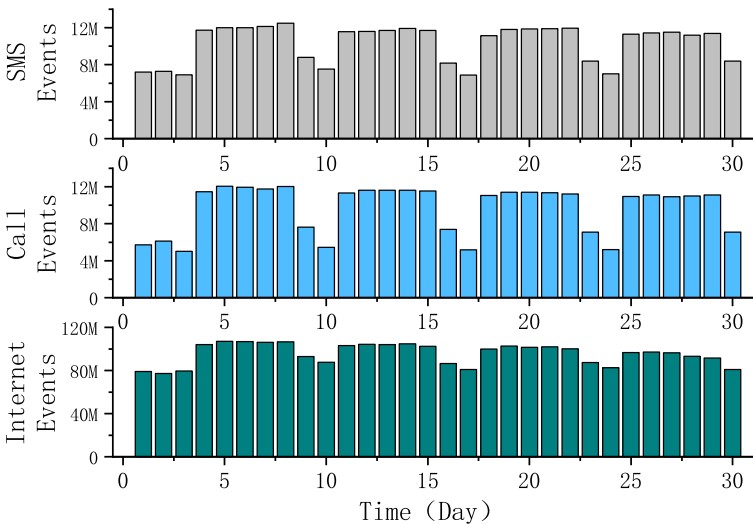

**Figure 2.** Daily traffic change statistics.

## 2.2. Spatial Domain

Figure 3 shows the spatial distribution of Call traffic over a certain period of time. We can easily find that the traffic distribution of the whole city was very uneven. Intensive traffic was concentrated in the downtown area, while urban suburbs had very sparse traffic distributions. Moreover, it can be clearly seen from Figure 3 that a few areas covered traffic much higher than other areas. These areas were usually the bustling areas of the city and were often the most burdensome areas for wireless networks, and accurate predictions of traffic in these areas are important. However, these areas carrying large amounts of traffic also pose great difficulties for traffic prediction modeling. The existence of such singular values was very detrimental to the fitting of the model. Moreover, the trend of traffic fluctuations in these areas was often much larger than in other areas, and more research is needed to capture its space-time characteristics well.

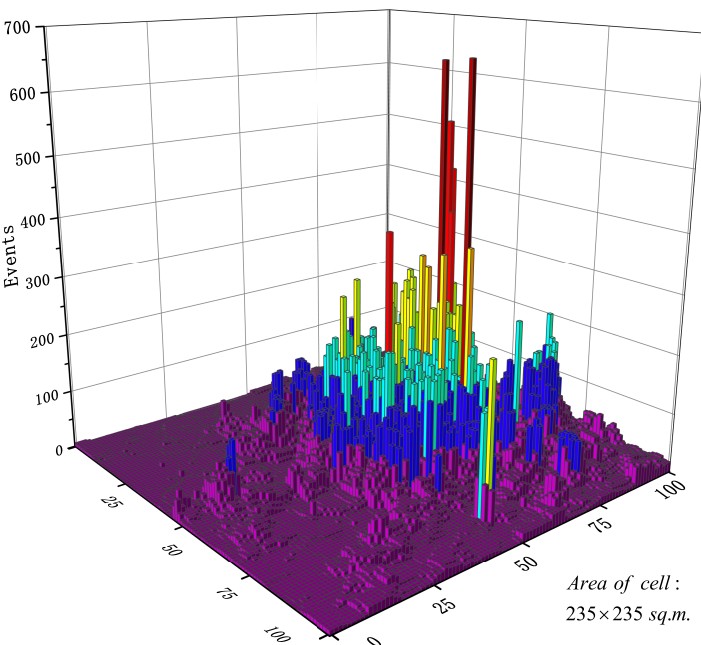

**Figure 3.** Spatial distribution of cellular network traffic.

The correlation between traffic changes in different cells cannot be seen intuitively through Figure 3. To better show the spatial correlation of cellular traffic, we extracted 11 × 11 cells in the Call

dataset for correlation analysis. Figure 4 shows the Pearson correlation coefficient $\rho$ for the spatial correlation between target cell $x^{(i,j)}$ and its neighboring cells $x^{(i',j')}$. The Pearson correlation coefficient is a widely used metric [16,17]. Its definition is expressed as follows:

$$\rho = \frac{\text{cov}\left(x^{(i,j)}, x^{(i',j')}\right)}{\sigma_{x^{(i,j)}}\sigma_{x^{(i',j')}}} \tag{2}$$

where *cov* represents the covariance operator and $\sigma$ represents the standard deviation. Figure 4 shows that the spatial correlation between different urban areas depended not only on the their proximity, but also on many external factors. For example, Cell (3,4) was the same distance from cell (4,3) to target cell (5,5), but the correlation coefficient was very different (0.35 and 0.96). Generally, the change in traffic was not necessarily highly related to neighboring cells and may also be strongly related to non-adjacent cells. However, the traditional convolution can only extract the information of neighboring cells. Therefore, we needed to find new ways to get potentially relevant information.

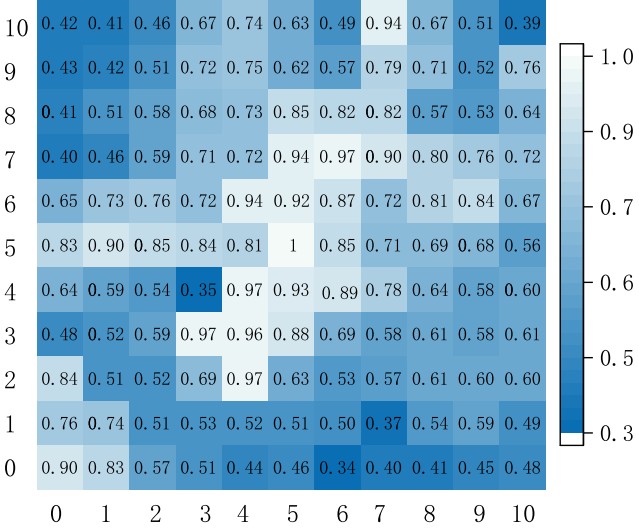

**Figure 4.** Spatial correlation analysis.

## 3. Cellular Traffic Prediction Model

### 3.1. Model Framework Introduction

This section mainly introduces our proposed hybrid spatiotemporal network HSTNet, which is mainly based on STDenseNet. HSTNet consists of two input sections and three module sections. The three modules include the convolution module, the time-embedding module, and the attention module. The predicted value $P'^{(h,w)}$ of the model was combined with the output data of the three modules. The model framework is shown in Figure 5.

The traffic of each time period in the datasets was counted by $100 \times 100$ cells; that is, the traffic data of each time period could be composed of a $100 \times 100$ traffic distribution matrix. Therefore, we could effectively extract the spatial correlation of cellular traffic through convolutional neural networks. The main prediction module of our model was also implemented by the improved solution of the convolutional neural network (DenseNet) [19]. The historical data included the last three time periods and the current time period of the previous three days. We considered that cellular traffic was not only highly correlated with the traffic distribution of the previous few hours, but also depended on the traffic distribution at the current moment of the previous few days. Therefore, we entered the two pieces of historical data into the model separately. For example, if we forecast the traffic at 12 o'clock on the 11th of a certain month, the input data were the traffic data of 12 h on 8/9/10 of the month and the traffic data of 9/10/11 o'clock on the day. After the historical data were entered

into the model, they were processed by two modules: the convolution module and the attention module. The convolution module consisted of two DenseNets with deformable convolutions [20] that handled the historical data of the two parts. The output matrix was fused through a matrix of learnable parameters.

In the second section, we analyzed the time correlation of cellular traffic data. There were strong correlations with cellular traffic at different times of the day and whether it was a holiday. The input data of the time embedding module included the hour value of the predicted time and the attribute of the date (working day or holiday). The matrix generated by the time embedding module would be added to the matrix output by the convolution module.

The attention module received the input of two historical data items and performed integration operations. Then, a weight matrix based on historical data was output. The matrix adjusted the weight of the output of the first two modules and output $O^{(h,w)}$. Finally, the final prediction matrix $P'^{(h,w)}$ was output through the sigmoid function and compared with the real value $P^{(h,w)}$.

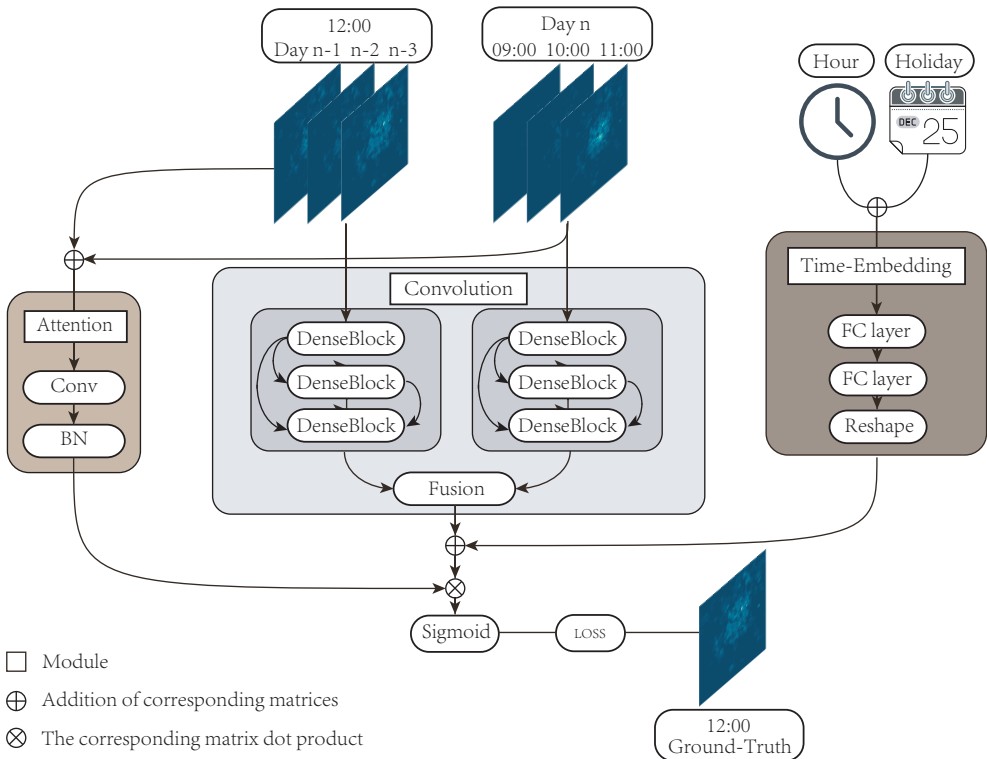

**Figure 5.** The hybrid spatiotemporal network's (HSTNet) framework structure.

## 3.2. Convolution Module

In recent years, the improvement of convolutional neural network performance has been mainly divided into two major areas. One area is depth, such as ResNet, which solves the problem of gradient disappearance when the network is too deep [21]. The other area is the width, such as the Inception network, which uses a multi-scale convolution kernel to extend the model's generalization capabilities. In STDenseNet, the prediction module consists of two densely concatenated convolutional networks with the same structure. Similar to ResNet, DenseNet also establishes a dense connection between the front and back layers; that is, each layer accepts the output of all the previous layers as input and implements feature reuse in turn. DenseNet relies on the ultimate use of the network architecture to achieve fewer parameters and leading performance compared to traditional models [19].

Cellular traffic has a strong spatiotemporal autocorrelation. This work uses DenseNet to extract the spatiotemporal features of historical data, which can achieve better results than the traditional single channel convolutional neural network. The convolution module in HSTNet contains two

separate DenseNets for processing two sets of historical data. Each DenseNet consists of three layers, each consisting of a unit block we call the DenseBlock. The outputs of the two DenseNets are multiplied by the learnable parameter matrix and then added.

### 3.3. Deformable Convolution

In recent years, convolutional neural networks have achieved excellent performance in many image fields with their good feature extraction ability and end-to-end learning. The convolution in the network samples different regions of the input image and then convolves the sampled information as an output. This convolution operation determines that the geometric deformation capability of the model does not come from the network, but from the diversity of the dataset. For example, assume that $Q$ represents the receptive field area covered by the convolution kernel, in a $3 \times 3$ convolution kernel, $Q = (1,1), (1,0), \ldots, (0,1), (1,1)$. For any pixel point $P_0$ on the feature map, the standard convolution method is as follows:

$$y(P_0) = \sum_{P_n \in Q} W(P_n) \cdot x(P_0 + P_n) \tag{3}$$

Because the ordinary convolution method has limited adaptability to the complex spatial correlation of cellular network traffic, we introduced a deformable convolution to the model. The process of deformable convolution is to add an offset variable $\Delta P_n$ at each sampling point position. $\Delta P_n$ can be continuously learned and adaptively changed according to the current image content. This means that the convolution kernel is not limited to a fixed position sampling method, but can search for the region of interest near the current position for sampling. Thus, the convolution kernel improves the feature extraction capability for complex spaces. The following is the calculation process of the deformable convolution:

$$y(P_0) = \sum_{P_n \in Q} W(P_n) \cdot x(P_0 + P_n + \Delta P_n) \tag{4}$$

The traditional convolution window only needs to train the pixel weight parameters of each convolution window. The deformable convolutional network must add some parameters to train the shape of the convolution window, that is the offset vector of each pixel. The offset field in Figure 6 is the additional parameter to be trained, and its size is the same as the input picture size. The convolution window slides on the offset field to achieve the effect of convolution pixel offset and optimize the sampling points.

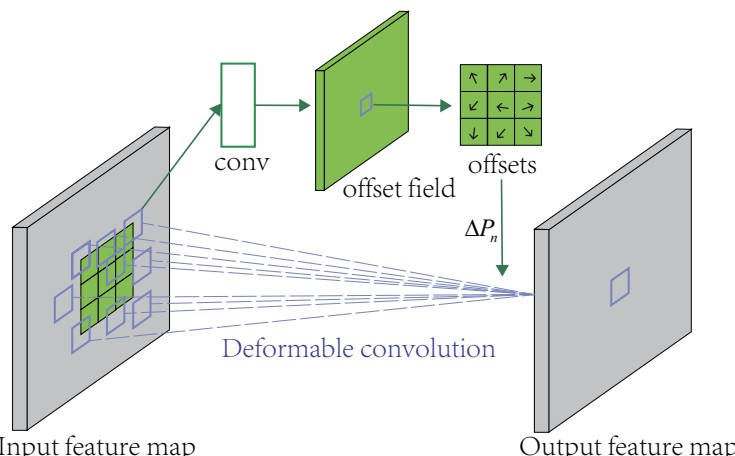

**Figure 6.** Illustration of a $3 \times 3$ deformable convolution.

It can be seen from the above analysis that if $\Delta P_n = 0$, the deformable convolution becomes a normal convolution, and there is no improvement in performance. In the process of predicting cellular

network traffic, an ordinary convolution can only extract features for a fixed size range. The deformable convolution can extend the feature extraction range to more effective areas around by learning the offset variable $\Delta P_n$. In this training process, the model can be offset from the area calculated by the common convolution kernel to other areas with more correlation and effectively avoids interference from uncorrelated spatial features, thus improving the predictive performance of cellular traffic.

In summary, to improve the spatial feature extraction ability of the model, a deformable convolution unit is added to each layer (DenseBlock) in the DenseNet. As shown in Figure 7, the original DenseBlock consists of a batch normalization layer, a rectified linear units (ReLU) layer, and a $3 \times 3$ convolution layer. The improvement of this paper is to change the $3 \times 3$ convolution layer to a $1 \times 1$ size, which is used to shape the data of the current DenseBlock, then access the batch normalization(BN) layer, the ReLU layer, and the deformable convolution layer in turn.

Since the traditional DenseBlock reuses the features of all previous layers, the deeper DenseBlock requires more parameters. Adding a $1 \times 1$ convolution layer can integrate input features into low dimensions and reduce the number of model parameters. We replaced the improved DenseBlock in the three-tier DenseNet, and the model parameters dropped from 230 thousand to 170 thousand. Replacing traditional convolutions with deformable convolution can expand the perceived range of convolution and improve the feature extraction capabilities of spatial features.

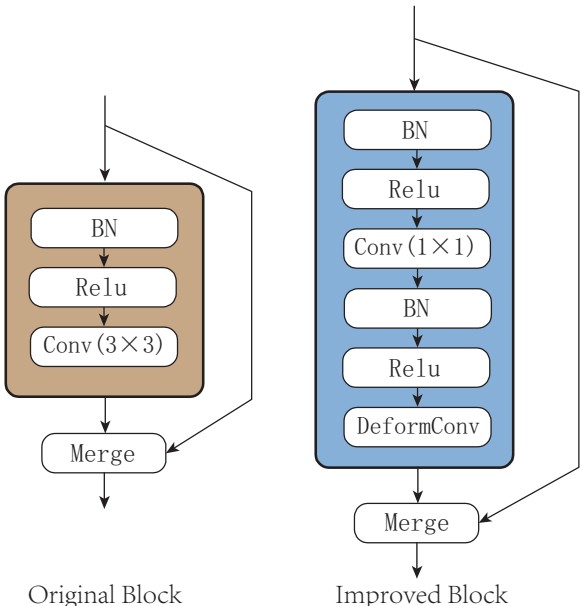

Original Block      Improved Block

**Figure 7.** Improved structure of DenseBlock.

*3.4. Time Embedding Module*

In the previous data analysis, we can see that time had a strong correlation with communication traffic. To capture the temporal characteristics of the data for better model prediction, we collected Italian holiday information and introduced hours and holidays as external features in the model input. The specific process was as follows.

- Dividing the time period of the day into 24 segments, representing 24 h, the time attribute of each data was represented by a 24-dimensional one-hot vector (*Hour_of_Day*).
- Holiday (including weekends and Italian festivals) is represented by a one-dimensional vector (*Is_of_Holiday*) and is entered with 0 or 1, 1 indicating that the day is a holiday and 0 indicating that the day is a working day.

The above two vectors were combined into a 25-dimensional vector *T*. For example, if the predicted time point is 12:00:00 11/01/2013 (Italian holiday), the two time data *Is_of_Holiday* (1) and *Hour_of_Day* (a 24-dimensional one-hot vector, its 12th bit is 1) are extracted to form a time feature

vector $T$. The feature vector $T$ is input to the two layer fully connected layer, and the output is an $H \times W$ dimensional vector $v_{time}$. The vector is reshaped into a matrix $M_{time}$ of $H \times W$ in size through a reshape layer and merged with the input result of the prediction branch. The reshape layer receives a vector of length $H \times W$ and shapes it into a matrix of $H \times W$. The calculation process is as follows:

$$v_{\text{time}} = \sigma \left( W_{\text{time}}^2 \sigma \left( W_{\text{time}}^1 T + b_{\text{time}}^1 \right) + b_{\text{time}}^2 \right) \tag{5}$$

where $W_{\text{time}}^i$ and $b_{\text{time}}^i$ are the learnable parameters of the i$^{\text{th}}$ fully connected layer. $\sigma$ represents the sigmoid activation function.

$$M_{time} = \text{Reshape} \left( v_{\text{time}} \right) \tag{6}$$

where $v_{\text{time}} \in R^{HW \times 1}$ and $M_{time} \in R^{H \times W}$.

### 3.5. Attention Module

The human brain receives considerable external input information at every moment. When the human brain receives this information, it consciously or unconsciously uses the attention mechanism to obtain more important information. At present, this attention mechanism has been introduced into the fields of natural language processing, object detection, semantic segmentation, etc., and has achieved good results. In our work, an attention mechanism was added to the network as a module to optimize the density map generated by the prediction.

The statistical value of the most densely populated area is often much larger than the value of most other areas in the traffic dataset. For example, in a traffic distribution matrix, the average is 30, but the maximum is 4000. This is very disadvantageous for accurate prediction of images by convolutional neural networks. Therefore, to solve the problem that the value gap between different regions is too large in the model prediction process and improve the overall prediction performance of the model, this paper proposes a weight adjustment scheme based on an attention mechanism. The density map traffic density distribution has a strong correlation with the corresponding historical data. Therefore, we integrated the corresponding historical data as input, merged them into a two-column attention matrix through a $1 \times 1$ convolution kernel, and normalized it to form a weight matrix $W^{(h,w)}$. Then, $W^{(h,w)}$ multiplies the matrix $M^{(h,w)}$ generated by the prediction branch to generate a prediction matrix $O^{(h,w)}$ for adjusting the weights. The calculation is as follows:

$$O^{(h,w)} = W^{(h,w)} \cdot M^{(h,w)} \tag{7}$$

In this way, higher weights can be obtained for pixels with relatively higher traffic density in historical data, and lower weights can be obtained for pixels with relatively lower density. In the case, weights are differentiated from the density map generated by the prediction module. The operation can improve the quality of the final density map.

## 4. Experimental Results and Analysis

### 4.1. Experimental Process and Parameter Setting

The experimental dataset was from Telecom Italia, and we used the same pre-processing method as [17] to aggregate data from the 10 min interval in the original dataset to hours. Because the 10 min interval dataset was quite sparse, it was not conducive to extracting spatiotemporal characteristics. The difference was that [17] separately predicted the receive and send dimensions in SMS and Call. We combined the receive and send dimensions and used the total traffic input model for prediction. All data were normalized before inputting in the model, which allowed the model to converge faster and improved the computational efficiency of the fitting process.

HSTNet uses an optimization algorithm, Adam [22], which can replace the traditional stochastic gradient descent algorithm, which iteratively updates the neural network weights based on the training

data. The experiment was carried out in three datasets with a learning rate of 0.01 and a training of 150 epochs. The learning rate decayed as the epoch of training increased. Our model was tested on three datasets: SMS, Call, and Internet. Each dataset contained 1488 (62 days $\times$ 24 h) slices. Except for the first three days without sufficient historical data, we used 52 days of data (1248 pieces) from 4 November 2013 to 24 December 2013 as the training set and used data from 25 December 2013 to 1 January 2014 (168 pieces) as the test set. In the convolution module, the deformable convolutional layer had 32 filters. The remaining convolutional layers had 16 filters with a kernel size of $1 \times 1$. The normal convolutional layer in the attention module had one filter with a kernel size of $1 \times 1$. The activation function of the convolutional layers was ReLU, except for the last layer, which used the sigmoid activation function. The code of HSTNet was implemented under Python 3.7, Keras 2.1.6, and NumPy 1.15.4. The experimental hardware environment included AMD R5 2600, GTX 1070, and 16 G memory.

Our experiments compared HSTNet performance with baseline algorithms such as the historical average (HA), ARIMA, LSTM, and STDenseNet. In the experiment process, the deformable convolution, time embedding module, and attention module were embedded in STDenseNet to observe the performance improvement of the model, and HSTNet had all the above improvements added. The generated prediction map was re-adjusted to the normal scale and then evaluated with the true value.

We used the two indicators of mean absolute error (MAE) and root mean squared error (RMSE) to evaluate the model. MAE is the average of the absolute error, which can better reflect the actual situation of the predicted value error.

$$MAE = \frac{\sum_{h=1}^{H}\sum_{W=1}^{W}\left| p'^{(h,w)} - p^{(h,w)} \right|}{H \times W} \tag{8}$$

RMSE represents the square root of the second sample moment of the differences between predicted values and observed values or the quadratic mean of these differences. RMSE is more sensitive to outliers.

$$RMSE = \sqrt{\frac{\sum_{h=1}^{H}\sum_{W=1}^{W}\left( p'^{(h,w)} - p^{(h,w)} \right)^2}{H \times W}} \tag{9}$$

*4.2. Experiment Analysis*

To compare the performance of HSTNet proposed in this paper, we selected three existing traffic prediction algorithms as the baseline of this experiment: historical average (HA), ARIMA, LSTM, and STDenseNet. Four methods performed MAE- and RMSE-based evaluations on three datasets. The result is shown in Figure 8.

From Figure 8, we can see that HSTNet's MSE and RMSE performance in the three traffic datasets was ahead of the other existing algorithms. The historical average was simply calculated from historical data and lacked the mining of deep correlations. ARIMA only considered the historical timing characteristics of the data, without regard for other dependencies. The performance of LSTM was better than statistical methods, but worse than other deep learning methods. STDenseNet did not consider the impact of external factors. Our model not only better extracted the spatial correlation of traffic data, but also considered the impact of time attributes on traffic changes, so the best performance was achieved.

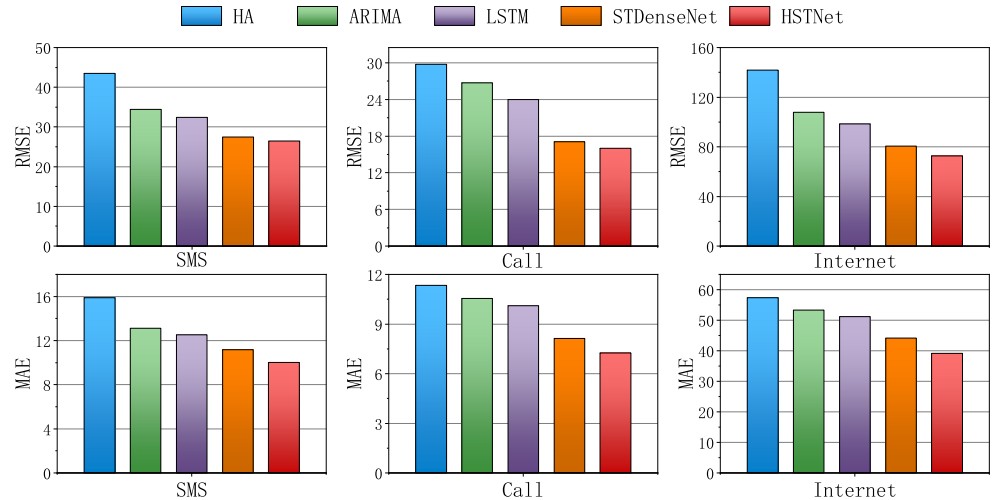

**Figure 8.** Comparison of prediction performance on the baseline and HSTNet. HA, historical average.

In Table 1, we calculate the model evaluation results after adding the deformable convolution, time embedding module, and attention module in STDenseNet. In Table 1, +DeformConv represents embedding only deformable convolutions in the baseline. HSTNet was evaluated after incorporating three improvements. The model achieved the best MAE and RMSE performance on Call, and the effect on SMS was slightly worse. The effect of the model on Internet improved, but the overall performance was far worse than Call and SMS because the traffic on Internet was very different from the SMS and the Call traffic. In some cases, it was close to ten times the gap. Considerable traffic changes had a large impact on model performance.

It is worth mentioning that the traffic gap between different cells after integration was about twice that of the original, which was more complicated than the separate prediction. Therefore, compared to the results in [17], our experiments obtained larger RMSE and MAE on SMS and Call.

**Table 1.** Overall performance of the model.

| Dataset | Model | MAE | RMSE |
|---------|-------|-----|------|
| SMS | STDenseNet | 11.10 | 27.49 |
| | +DeformConv | 10.81 | 26.91 |
| | +Time-property | 10.66 | 27.22 |
| | +Attention | 10.09 | 26.62 |
| | HSTNet | 10.01 | 26.42 |
| Call | STDenseNet | 8.13 | 17.10 |
| | +DeformConv | 7.61 | 16.18 |
| | +Time-property | 8.03 | 16.89 |
| | +Attention | 7.27 | 16.70 |
| | HSTNet | 7.25 | 16.04 |
| Internet | STDenseNet | 44.15 | 80.51 |
| | +DeformConv | 43.23 | 77.75 |
| | +Time-property | 39.73 | 77.08 |
| | +Attention | 39.89 | 74.48 |
| | HSTNet | 39.19 | 72.72 |

Figure 9 shows the performance improvement of the model by adding different modules and HSTNet. For the SMS datasets, the addition of deformable convolution, time embedding, and attention modules increased by 2.61%, 3.96%, and 9.09%, respectively, the RMSE. For the MAE, there were 2.11%, 0.98%, and 3.16% performance improvements. For the Call dataset, the addition of deformable convolution units had an increase in 6.39% and 5.38% in MAE and RMSE, respectively; the attention module had a 10.57% improvement in MAE. For the Internet dataset, the three improvements we

proposed still improved the results to varying degrees. The effect of adding the time attribute and the attention module was very obvious and had an approximately 10% improvement on the MAE.

It can be seen that the three different improvements improved the results in the different datasets, verifying the correctness of the hypotheses in Section 3. Different improvements had different effects on the three datasets. The attention module was very impressive for predicting the performance of all datasets. Deformable convolution and time embedding modules also had varying degrees of performance improvement for individual datasets. Overall, the performance improvement created by the attention mechanism was greater than the other two improvements.

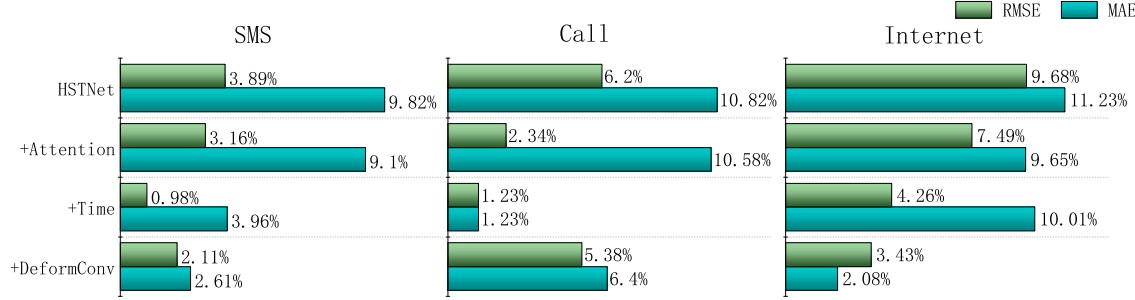

**Figure 9.** Comparison of different module effects based on STDenseNet.

Embedding different modules had different effects on the model's runtime and parameters. Table 2 shows the changes in the time to train an epoch and the number of parameters under different conditions. We added each module individually to observe the results. Although the time embedding module and the attention module had added parameters, they had little effect on the running time. The addition of deformable convolution reduced the parameters by about 69K, but complex calculations increased the running time by nearly half. There was no difference in the running cost of the three datasets due to the same structure.

Considering the performance and cost of HSTNet, we think that the time embedding module and attention module were the most effective strategies for improving model performance. Deformable convolution could improve prediction performance to a certain extent, but it also increased the running time of the model. It should be mentioned that DenseBlock with more filters could obtain better performance, but the training time was greatly increased. For example, if we replaced the original 32 filters with the 64 filters, we could increase the above results of RMSE by 2.3%, but nearly three times the training time in SMS.

**Table 2.** The effects on parameters and the time of each epoch.

| Model | Time | Parameters |
|---|---|---|
| STDenseNet | 22s | 239K |
| +DeformConv | 34s | 170K |
| +Time-property | 23s | 350K |
| +Attention | 22s | 243K |
| HSTNet | 35s | 284K |

In Section 3.1, we mentioned that the input data included the last three time periods and the current time period of the previous three days. We input different time dimensions to analyze the impact on HSTNet's performance. The $N$-dimensional data represented the data from the last $N$ time periods and the current time period of the previous $N$ days. As shown in Table 3, different time dimensions had a certain impact on the RMSE results. In the three datasets, HSTNet achieved the best performance when inputting three-dimensional data. The performance was slightly poorer when only one- or two-dimensional data were input, which indicated that the model could not extract the

spatiotemporal characteristics of the traffic well in this case. When $N$ = 4, the performance of the model would also decrease. We believe that this was caused by the introduction of more low correlation data.

**Table 3.** RMSE results of different input dimensions.

| Input Dimension | 1 | 2 | 3 | 4 |
|---|---|---|---|---|
| SMS | 27.51 | 27.18 | **26.42** | 26.83 |
| Call | 16.86 | 16.23 | **16.04** | 16.62 |
| Internet | 80.10 | 75.38 | **72.72** | 78.32 |

Figure 10 shows the cell of the Internet dataset (55,58), the predicted values of the five methods compared to the real values. To visually show the difference in performance, we compared their accumulation error in the lower right subgraph of Figure 10. The predicted values of HA and ARIMA had large errors with the true values, and they lacked accuracy in fitting the peaks. The method based on deep learning was much better than the traditional method. HSTNet's overall fitting effect on traffic was higher than the other three. Especially during peak hours of network traffic, HSTNet achieved more accurate predictions. Compared to STDenseNet, its overall error in this area decreased by nearly 10% because HSTNet had better spatiotemporal feature extraction capabilities.

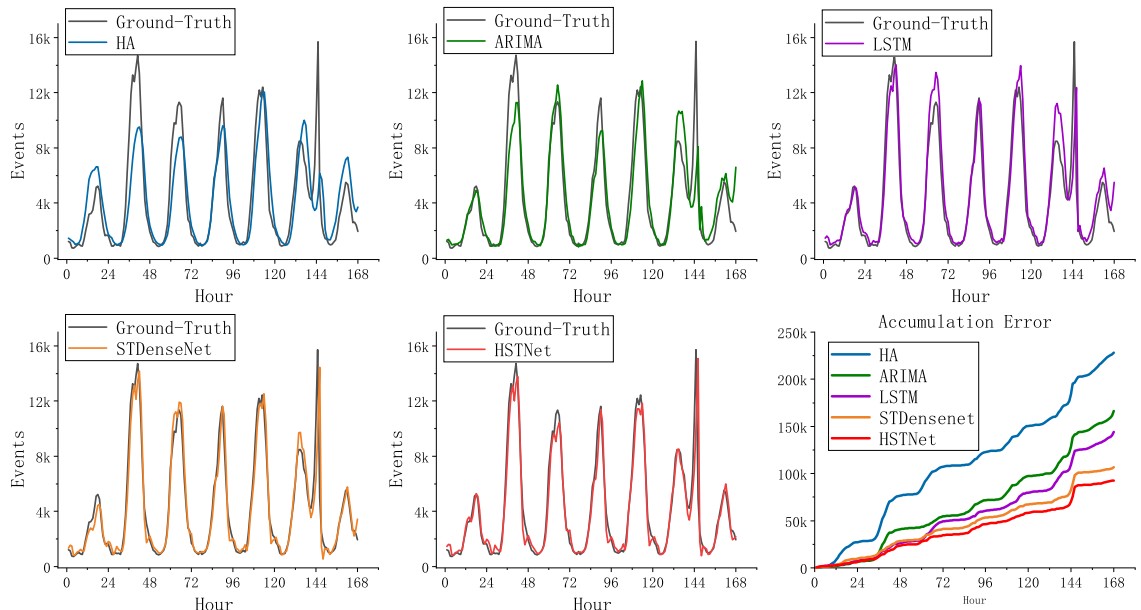

**Figure 10.** Comparison of the baseline and HSTNet on cell (50,58).

### 4.3. Experimental Result

In this part, we use HSTNet to predict the dataset and analyze the results. To verify the predictive performance of our model, we compared the predicted and actual values of the city's total traffic over one week and performed error analysis.

As shown in Figure 11, the X-axis represents the time interval in hours, and the Y-axis represents the total traffic for the entire city at the current time. Even for the difficult task of predicting overall urban traffic, the model still had a good fitting effect. A sharp rise in traffic was observed at 145 h in the figure due to the large error caused by the 2013 New Year's Eve event. The ability of the model to fit the traffic fluctuations caused by unexpected events needs to be improved.

HSTNet had the best prediction effect on the Call dataset. From the top right subgraph of Figure 11, it can be seen that the error value could be controlled within a certain range except for the impact caused by the unexpected event. The prediction effect of Internet traffic at night was not very

satisfactory. There was a significant error in traffic prediction during this period, as shown in the lower right subgraph, because the three types of traffic had different scales of change. The Internet's day-night traffic gap was approximately two million (SMS and Call were only approximately 500 k), which created considerable difficulties in forecasting.

HSTNet also had good predictive performance for traffic in different areas of the city. Figure 12 shows a comparison of the predicted and real images at 10 o'clock on 24 December 2013. Each image had 100 × 100 cells. The brightness of the cell pixels indicates the traffic load of the corresponding cell. The brighter area indicates greater traffic load. This proved that our model could accurately predict the regions with different traffic distributions and could extract the spatial correlation of urban cellular traffic.

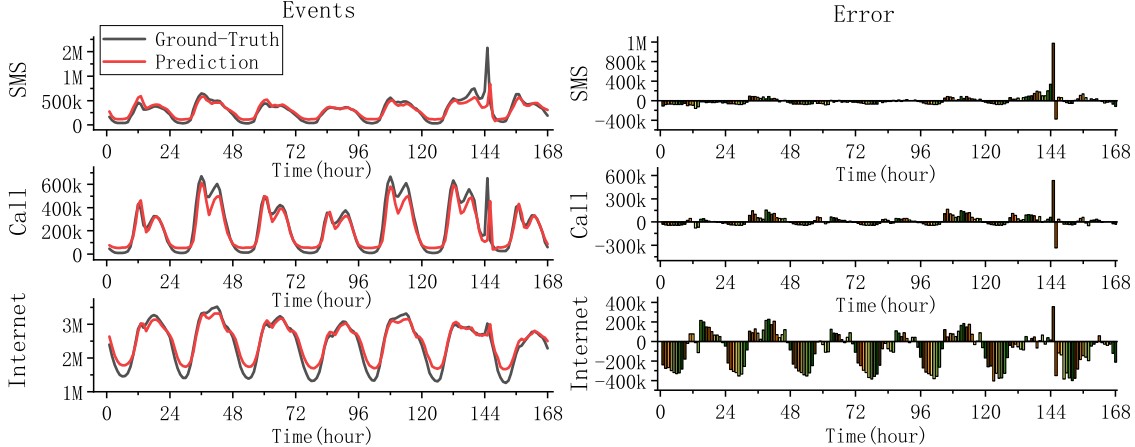

**Figure 11.** Comparison of hourly traffic prediction with the ground truth.

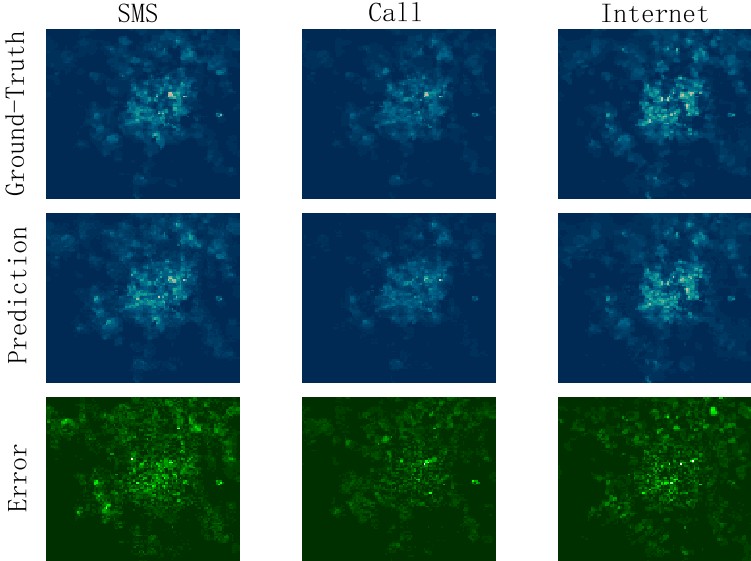

**Figure 12.** Comparison of predicted and real images.

The last line in Figure 12 shows the error between the corresponding prediction map and the ground truth. The brighter area indicates a greater error. We can see that the prediction error was not only biased towards the downtown area where the traffic load was large, but also, there were large errors in many suburbs with heavy traffic loads. This shows that the change in cellular traffic also depended on many factors that we had not considered, and a more sophisticated model is needed in the future to analyze the characteristics of cellular traffic changes.

## 5. Conclusions

This paper was devoted to the prediction of cellular network traffic. We conducted in-depth research on the spatiotemporal correlation of cellular networks and analyzed various factors that affect traffic changes, then proposed a hybrid deep learning model for traffic spatiotemporal prediction:

- This work used DenseNet with deformable convolution to extract the spatiotemporal characteristics of traffic.
- We introduced hour and holiday information to aid traffic forecasting.
- We proposed an attention module based on historical data to adjust the weight of the predicted traffic.

The experimental results showed that compared with the existing methods, the hybrid spatiotemporal network HSTNet proposed in this paper could better extract the spatiotemporal characteristics of image traffic data and improve the prediction accuracy, thus making more effective traffic prediction.

There are still many aspects that need improvement in our work:

- The model did not have a good ability to respond to fluctuations caused by emergencies.
- The forecast performance of the large scale traffic volume (total traffic volume of the entire city) needs to be improved.
- There are many external factors that we did not consider that could have a potential impact on cellular traffic changes.

In future work, traffic prediction modeling needs to consider not only the use of more sophisticated networks to extract features, but also the analysis and introduction of external data in multiple dimensions. It is worth mentioning that the introduction of our cellular traffic prediction scheme into traffic prediction problems in other similar contexts is also a very worthwhile research direction.

**Author Contributions:** Conceptualization, D.Z.; data curation, B.Y.; formal analysis, L.L.; funding acquisition, D.Z.; writing, original draft, D.Z. and L.L.; writing, review and editing, C.X. and Q.L. All authors have read and agreed to the published version of the manuscript.

**Funding:** This research was funded by the National Natural Science Foundation of China Grant Numbers 61402397, 61263043, 61562093, and 61663046.

**Acknowledgments:** This work is supported by: (i) the Natural Science Foundation China (NSFC) under Grant Nos. 61402397, 61263043, 61562093, and 61663046; (ii) Yunnan Provincial Young academic and technical leaders reserve talents under Grant No. 2017HB005; (iii) the Yunnan Provincial Innovation Team under Grant No. 2017HC012; and (iv) the Youth Talents Project of the China Association of Science and Technology under Grant No. W8193209.

**Conflicts of Interest:** The authors declare no conflict of interest.

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
