# Peer review of "Citywide Cellular Traffic Prediction Based on a Hybrid Spatiotemporal Network"

_algorithms, doi:10.3390/a13010020_

Round 1

Reviewer 1 Report

This work uses deep learning to make traffic predictions based on call, sms and internet traffic data. Although this is a hot topic and the authors approach use DenseNet with deformable convolution, it feels like this is mostly a work that builds severly on previous results and basic theory with limited original scientific contribution. Some comments on the current research and presentation are as follows.

Typos and syntax comments:

pg2: ln40, ln50, ln53-54 pg3: ln71-72

Clarifications:

what does it mean to combine SMS and Call data into one? What is that you are combining exactly and how? (e.g., aggregating number of calls and sms's?). I feel that the data description part should be improved to clearly state the type of data considered, the amount, the spatial areas and the temporal periods. For example, for the spatial area considered, just mentioning the number of rows x columns you chose to make the grid says nothing about the granularity of the data on this domain.  This also applies, and is even more important, in the description of the data used in the experiments section. What is an epoch? What do you mean by length of a dataset? etc etc. For Fig.1 it would be nice if x-axis showed the actual hour of the day each bar corresponds to. For Fig 3, a random reader cannot appreciate the result, only to believe that the red bars correspond to Milan downtown, but also, as commented before, we need to understand what size of area each bar considers. When talking about distance of cells in subsection 2.2, you need to first define that distance (what do you consider and how do you measure it). Also, it is hard to appreciate the purpose of the spatial correlation analysis from this very simplistic approach. Firstly, one could easily argue about the use of pearson's correlation, but also, authors need to give better motivation on what they are attempting to show and why showing it through correlation makes sense. The choice of temporal input a tthe model description is not clear or persuasive. Why only 3 hours and 3 days for example and not 4? How does that arbitrary choice affect results, or is it arbitrary at all?

Additional material:

It would be nice to have an evaluation of the performance of the system in general, in terms of training time etc, not just its prediction performance.

Author Response

Typos and syntax comments:

pg2: ln40, ln50, ln53-54 pg3: ln71-72

Our response: Thank you for your comments. We have revised the above problems.

what does it mean to combine SMS and Call data into one? What is that you are combining exactly and how? (e.g., aggregating number of calls and sms's?).

Our response: Thank you for your comments. Our description is not clear enough. In SMS and Call datasets, we combine the traffic of the receiving and sending dimensions into one but not SMS and Calls. So, we can better compare the spatiotemporal characteristics of three datasets. We have modified the corresponding description at line 75-77 and Line 263-264.

I feel that the data description part should be improved to clearly state the type of data considered, the amount, the spatial areas and the temporal periods. For example, for the spatial area considered, just mentioning the number of rows x columns you chose to make the grid says nothing about the granularity of the data on this domain. This also applies, and is even more important, in the description of the data used in the experiments section.

Our response: Thank you for your comments. In order to more clearly describe the data information, we have modified and added descriptions in Line 70-77, line121-127, line263-264 and line 271-273. We updated the coordinates in Fig 1, Fig 4 and Fig 11. We added the cell area in Fig 3.

What is an epoch? And what do you mean by length of a dataset? etc etc.

Our response: Thank you for your comments. An epoch refers to the process by which all training data is sent to the network for a single forward computation and back propagation.

Traffic dataset is recorded from 00:00 11/01/2013 to 23:59 01/01/2014. We integrate the data into hours. Therefore, the length of the pre-processed dataset is 1488 (62 day * 24 hour). We added the corresponding description at line 73 and line 271-274.

For Fig.1 it would be nice if x-axis showed the actual hour of the day each bar corresponds to.

Our response: Thank you for your comments. We have modified the x-axis of Figure 1.

For Fig 3, a random reader cannot appreciate the result, only to believe that the red bars correspond to Milan downtown, but also, as commented before, we need to understand what size of area each bar considers.

Our response: Thank you for your comments. In Fig3, the color of the flow bars in different cells only represents different degrees of statistical value. According to the paper [18] of datasets, the area of Milan is composed of a grid overlay of 10000 cells with size of about 235 × 235 meters. We have added description at line 70-72 and Figure 3.

When talking about distance of cells in subsection 2.2, you need to first define that distance (what do you consider and how do you measure it). Also, it is hard to appreciate the purpose of the spatial correlation analysis from this very simplistic approach. Firstly, one could easily argue about the use of pearson's correlation, but also, authors need to give better motivation on what they are attempting to show and why showing it through correlation makes sense.

Our response: Thank you for your comments. The correlation between traffic changes in different regions cannot be seen intuitively through Figure 3. So we introduce the Pearson correlation to analyze the correlation degree of traffic in different regions and extract an 11*11 cells in the SMS dataset for correlation analysis. The size of each cell in Figure 4 is equivalent to the cell in Figure 3. We have added coordinates in Figure 4.

It can be seen from Figure 4 that the change in traffic is not necessarily highly related to neighboring cells, and may also be strongly related to non-adjacent cells. However, the traditional convolution can only extract the information of neighboring cells, So we need to find new ways to get potential relevant information. We have added corresponding descriptions in line116-118 and line 121-127.

The choice of temporal input a the model description is not clear or persuasive. Why only 3 hours and 3 days for example and not 4? How does that arbitrary choice affect results, or is it arbitrary at all?

Our response: Thank you for your comments. The current input dimensions enable optimal performance of the model. We added comparative experiments in subsection 4.2 (line 336-344). It is verified that 3 periods of data can help the model achieve optimal performance.

Additional material:

It would be nice to have an evaluation of the performance of the system in general, in terms of training time etc, not just its prediction performance.

Our response: Thank you for your comments. We have added the analysis of training time and parameters in the subsection 4.2 (line 322-335).

Reviewer 2 Report

The manuscript considers a deep-learning framework for cellular traffic prediction based on spatiotemporal features. I have the following comments that need to be addressed :

1- In line 179, the deformable convolution is not clear, I recommend adding a sample figure of the deformable convolution kernels to show clearly the differences with the conventional one

2- In section 4.1, the authors have combined the datasets of SMS and Calls in one  set, What is the justification of such assumption especially that both have different characteristics

3- By investigating most of the figures and results, The improvement compared to STDenseNet is not big. What is the cost in complexity of achieving this small improvement

4- In line 280, the author mentioned " It should be mentioned that HSTNet with a wider DenserBlock can obtain better predictions, but the training time is greatly increased", kindly add a quantitative measure of this observation

5- The language of the paper needs to be improved, such as the following:

 i)  Line 40, the following sentence is not clear "For the first method includes autoregressive integrated moving average (ARIMA)", kindly rephrase.

ii) Line 93, 94, check the sentence "... , and the traffic on holidays is much lower than that on holidays, which ..."

Author Response

1- In line 179, the deformable convolution is not clear, I recommend adding a sample figure of the deformable convolution kernels to show clearly the differences with the conventional one.

Our response: Thank you for your comments. We have added figure 6 and corresponding description in subsection 3.3 (line 191-195).

2- In section 4.1, the authors have combined the datasets of SMS and Calls in one set, What is the justification of such assumption especially that both have different characteristics.

Our response: Thank you for your comments. Our description is not clear enough. In SMS and Call datasets, we combine the traffic of the receiving and sending dimensions into one but not SMS and Calls. So, we can better compare the spatiotemporal characteristics of three datasets. We added the corresponding description at line 75-77 and Line 263-264.

3- By investigating most of the figures and results, The improvement compared to STDenseNet is not big. What is the cost in complexity of achieving this small improvement.

Our response: Thank you for your comments. We have added an analysis of model costs in the subsection 4.2 (line 322-355).

4- In line 280, the author mentioned " It should be mentioned that HSTNet with a wider DenserBlock can obtain better predictions, but the training time is greatly increased", kindly add a quantitative measure of this observation.

Our response: Thank you for your comments. For example, if we replace the original 32 filters with the 64 filters, we can increase the above results of RMSE by 2.3% but nearly three times of training time in the SMS. We have added corresponding descriptions in line 333-335.

5- The language of the paper needs to be improved, such as the following:

i) Line 40, the following sentence is not clear "For the first method includes autoregressive integrated moving average (ARIMA)", kindly rephrase. ii) Line 93, 94, check the sentence "... , and the traffic on holidays is much lower than that on holidays, which ..."

Our response: Thank you for your comments. We have modified these issues in line 40 and line 95-96.

Round 2

Reviewer 1 Report

I would like to thank the authors for taking into consideration the comments of the reviewers and significantly improving the manuscript. Although there is always room for improvement, I feel that the current shape of the paper is sufficient for publication.

Reviewer 2 Report

Thank you for addressing the reviewers' comments. I have no further comments